# A Tissue Paper/Hydrogel Composite Light-Responsive Biomimetic Actuator Fabricated by In Situ Polymerization

**DOI:** 10.3390/polym14245454

**Published:** 2022-12-13

**Authors:** Qijun Wu, Chao Ma, Lian Chen, Ye Sun, Xianshuo Wei, Chunxin Ma, Hongliang Zhao, Xiuling Yang, Xiaofan Ma, Chunmei Zhang, Gaigai Duan

**Affiliations:** 1Jiangsu Co-Innovation Center of Efficient Processing and Utilization of Forest Resources, International Innovation Center for Forest Chemicals and Materials, College of Materials Science and Engineering, Nanjing Forestry University, Nanjing 210037, China; 2State Key Laboratory of Marine Resource Utilization in South China Sea, Hainan University, Haikou 570228, China; 3Key Laboratory of Quality Safe Evaluation and Research of Degradable Material for State Market Regulation, Products Quality Supervision and Testing Institute of Hainan Province, Haikou 570203, China; 4Institute of Materials Science and Devices, School of Materials Science and Engineering, Suzhou University of Science and Technology, Suzhou 215009, China

**Keywords:** biomimetic composite actuator, poly(*N*-isopropylacrylamide) (PNIPAM) hydrogel, tissue paper, in situ polymerization, photothermal responsiveness

## Abstract

Stimulus-responsive hydrogels are an important member of smart materials owing to their reversibility, soft/wet properties, and biocompatibility, which have a wide range of applications in the field of intelligent actuations. However, poor mechanical property and complicated fabrication process limit their further applications. Herein, we report a light-responsive tissue paper/hydrogel composite actuator which was developed by combining inkjet-printed tissue paper with poly(N-isopropylacrylamide) (PNIPAM) hydrogel through simple in situ polymerization. Due to the high strength of natural tissue paper and the strong interaction within the interface of the bilayer structure, the mechanical property of the composite actuator was highly enhanced, reaching 1.2 MPa of tensile strength. Furthermore, the light-responsive actuation of remote manipulation can be achieved because of the stamping graphite with high efficiency of photothermal conversion. Most importantly, we also made a few remotely controlled biomimetic actuating devices based on the near-infrared (NIR) light response of this composite actuator. This work provides a simple strategy for the construction of biomimetic anisotropic actuators and will inspire the exploration of new intelligent materials.

## 1. Introduction

Stimulus-responsive hydrogels have become an ideal material for smart actuators because of their soft and wet properties, biocompatibility, and large deformation [1,2,3,4,5,6,7]. Hydrogel actuators can convert a variety of external stimuli (such as light [8,9,10], heat [11,12], electricity [13,14], magnetism [15,16], humidity [17,18], pH [19,20,21], etc.) into mechanical energy and produce controllable and reversible deformation. Therefore, they have been applied in many fields, such as software robots [22,23], intelligent grippers [24,25], valves [26], drug release [27,28,29], etc. However, traditional hydrogel actuators usually possess isotropic structure [30,31], which is still a great challenge for the reported actuators. Recently, many related research achievements have been obtained by magnetic field/electric field induction [32], directional freezing [33,34], and self-assembly [35], and the bilayer structure as one of the most simple and common anisotropic construction methods has been widely used, but the complex preparation process is still an urgent problem to be solved. 

Regarding the near-infrared (NIR) light-responsive hydrogel actuator, because of its adjustable light intensity and irradiation position, the driving speed and shape of the actuator can be changed arbitrarily beyond remote control, which has further attracted people’s attention [36]. At present, NIR light-responsive hydrogel actuators are generally prepared by combining materials with photothermal conversion properties, such as graphene oxide [37], MXene [38], carbon nanotubes (CNTs) [39], Fe_3_O_4_ nanoparticles [40], and hydrogels, but these nanomaterials are expensive and do not have practical significance in large-scale use. Inkjet printing is a non-contact micron-level printing process, which can directly form black films [41]. It is considered to be a reliable printing process.

Poly(*N*-isopropylacrylamide) (PNIPAM) is the most typical thermosensitive hydrogel [42,43]. Its lower critical solution temperature (LCST) is close to 34 °C, and the hydrophilicity/hydrophobicity of PNIPAM changes around it, which is manifested in the shrinkage/expansion of volume. Since PNIPAM’s thermal phase transition behavior was discovered, it has been widely studied and regarded as an excellent thermal response material to be used in the preparation of hydrogel actuators. Tissue paper made from natural wood pulp is a low-cost, lightweight, deformable, and biocompatible material [44,45]. Lu et al. prepared Fe_3_O_4_/paper nanocomposites by a low-cost blending method, and then constructed a novel paper-based actuator with different shapes, showing good deformation, high stability, and reversible response [46]. The high mechanical strength ensures the potential of actuators to produce favorable output stiffness [47]. Moreover, paper quality can be easily modified. These reasons mean actuators built from tissue paper have attracted great attention.

Here, we report a tissue paper/PNIPAM hydrogel composite actuator, which can achieve near-infrared remote control. Graphite microparticles can evenly and densely cover the surface of tissue paper to provide high NIR light-adsorbing efficiency via a simple inkjet printer, owing to the super-hydrophilicity and surface roughness of the cellulose fiber-based tissue paper (Figure 1). In addition, we select the tissue paper because of not only its super-hydrophilic and rough/porous structure, but also its relatively robust, low-cost, and renewable properties. The convenient ultraviolet (UV) in situ polymerization method successfully combines tissue paper with hydrogel, so that the composite actuator can still maintain good performance after repeated recycling. As a result, this composite actuator with a firmly bonded anisotropic bilayer structure can provide various light-responsive deformations, because of the photothermal-responsive deswelling behavior of the PNIPAM hydrogel layer which can generate internal stress between the hydrogel layer and the tissue paper [36]. Furthermore, by imitating the design of the eagle’s claw when lifting and pulling heavy objects, the driving performance of the actuator is fully demonstrated. We believe that the tissue paper/PNIPAM hydrogel composite actuator uses eco-friendly and low-cost materials and a simple preparation process to achieve remote and flexible control. It has excellent application potential in the field of intelligent devices and provides a new inspiration for the construction of biomimetic actuators.

## 2. Experimental Section

### 2.1. Materials

Tissue paper was purchased from Golden Red Leaf Paper Group Co., Ltd. (Suzhou, China). *N*,*N*′-Methylenebis(acrylamide) (BIS, M_W_ = 154.17, 99%) was obtained from Aladdin Chemistry Co., Ltd. (Shanghai, China). *N*-isopropylacrylamide (NIPAM, M_W_ = 113.16, 99%) and 2,2′-Azobis(2- methylpropionamidine) dihydrochloride (AIBA, M_W_ = 271.19, 98%) were purchased from J&K Scientific Co., Ltd. (Beijing, China). Reactive red was purchased from Xingtai Wenyuan Chemical Dyes Tuff Co., Ltd. (Xingtai, China). Deionized water was produced by the laboratory. All the chemical reagents were used as received. 

### 2.2. Instruments

A printer (HP LaserJet MFP M227-M231), 365 nm UV light source (20 W), scanning electron microscope (Phenom Pro, Eindhoven, The Netherlands), mechanical tensile machine (UTM2502, SUNS, Zhangzhou, China), differential scanning calorimeter (DSC200F3, NETZSCH, Berlin, Germany), 808 nm NIR light source (VLSM-808-B-8-FA, Shanghai Hanyu Optical Fiber Communication Technology Co., Ltd., Shanghai, China), infrared thermal imaging instrument (Testo 872), spectrophotometer (Lambda 950, PE Ltd., Singapore) were used. All the physical photos of the paper were obtained by an iPhone 12 Pro., Cupertino, CA, USA.

### 2.3. Fabrication of the Tissue Paper/PNIPAM Hydrogel Composite Actuator

Firstly, the original white tissue paper was placed in an inkjet printer, and the tissue paper was printed twice on both sides to prepare the black tissue paper. The tissue paper/PNIPAM hydrogel composite actuator with 250 μm thickness (50 μm thickness of black tissue layer and 200 μm thickness of hydrogel layer) was prepared simply by in situ polymerization. NIPAM (monomer, 450 mg), *N*,*N*′- Methylenebis (acrylamide) (BIS) (crosslinker, 30 mg), 2,2′-Azobis(2- methylpropionamidine) dihydrochloride (AIBA) (photoinitiator, 18 mg) were dissolved in 4.5 ml deionized water to prepare prepolymer solution. First, the black tissue paper was fully impregnated with NIPAM solution. Second, a pipette gun was used to pour the NIPAM prepolymer into a self-made glass silicone rubber mold, and the tissue paper impregnated with the NIPAM prepolymer was placed into the mold. Next, the top of the mold was covered with another glass plate so that the solution did not leak out. Finally, the entire mold was exposed to UV light for 4 minutes to complete the polymerization. The tissue paper/PNIPAM hydrogel composite actuator was immersed in deionized water for 1 h to remove unpolymerized monomers. In follow-up experiments, the tissue paper/PNIPAM hydrogel composite actuator was cut into strips of different sizes before use.

## 3. Results and discussion

### 3.1. Microstructure and Morphology of the Tissue Paper/PNIPAM Hydrogel Composite Actuator

Figure 2 shows the microstructure characterization of the tissue paper/PNIPAM hydrogel composite actuator. As shown in Figure 2a,b, the easily available and environmentally friendly tissue paper with anisotropic properties provided the directional bending of the hydrogel composite actuator. Figure 2c,d fully show that after the application of an inkjet printer, the obtained black tissue paper was more uniformly covered by graphite microparticles owing to the super-hydrophilic and porous structure of the cellulose-based tissue paper, which not only can make the actuator show good photothermal conversion performance, but also can slightly improve the mechanical performance. A noteworthy problem was the combination between the active and inert layers of the bilayer actuator. It can be seen in Figure 2e,f that, after soaking the tissue paper in the prepolymerization solution for half an hour and then in situ polymerization of the NIPAM monomers, the combination of tissue paper and PNIPAM hydrogel layer was firm, on account of the super-hydrophilic and porous natural cellulose fibers of the tissue paper which can let the water and monomers penetrate into the tissue paper easily. In other words, some of the NIPAM monomers can be in situ polymerized in the tissue paper which can integrate the tissue paper and the PNIPAM hydrogel layer together.

### 3.2. Characteristics and the Actuating Properties of the Tissue Paper/PNIPAM Hydrogel Composite Actuator

The weaker mechanical property seriously limited the development of hydrogel actuators. As shown in Appendix A, the tensile strength of pure PNIPAM hydrogel was only about 27 kPa. However, the mechanical performance of the tissue paper/hydrogel composite actuator was highly enhanced, reaching about 1.2 MPa (Figure 3a), on account of the combined robust tissue paper (Appendix A), which can still maintain a high strength after being stamped by printing graphite. In contrast to previous reports, such as Al-alginate/Poly(*N*-isopropylamide) hydrogel (120 KPa) [48], PNIPAM/clay/CMCTs/GP hydrogel (137.9 kPa) [49], wood-PNIPAM composite hydrogel (1.1 MPa) [50], the tissue paper/PNIPAM hydrogel composite actuator has great advantages. It was significantly proved that the combination of tissue paper and PNIPAM hydrogel enhanced its tensile strength and helped its application in more fields. However, the tensile strength (1.2 MPa) of this composite actuator is not very high in existing hydrogel-based actuators. Therefore, to enhance the mechanical strength of this bilayer actuator in future, some applications with high-strength cellulose-based porous 2D materials (such as cotton clothes) should be attempted and the mechanical property of the stimulus-responsive deformational hydrogel should also be improved (such as designing a double-network structure).

As a black material, graphite microparticles have favorable light absorption which can confer good photothermal responsiveness to the tissue paper/PNIPAM hydrogel composite actuator. To confirm this, we tested the absorption of tissue paper and black tissue paper in the wavelength range of 400–950 nm by a UV–Vis spectrophotometer, and the results are shown in Figure 3b. For original white tissue paper, its transmittance in the NIR band was about 63%, which showed good light wave transmittance, while the transmittance of the black tissue paper uniformly covered by graphite in the NIR band can be sharply reduced to only 5%, which fully illustrated that the black tissue paper had good absorption of NIR light and provided a possibility for remote light control. 

In order to further explore the deformation of PNIPAM hydrogel, we studied its temperature sensitivity. As shown in Figure 3c, we recorded the weight of PNIPAM hydrogel at different temperatures, and calculated its equilibrium swelling rate. It can be seen that its equilibrium swelling rate decreased with the increase in temperature and dropped rapidly in the range from 25 to 35 °C. It showed that the hydrogel had a large deformation degree within this temperature range, which was consistent with the LCST results measured previously. The PNIPAM hydrogel sheet and black tissue paper (0.3 × 20 × 20 mm^3^) with the same size were placed in water at 5 °C to compare their size change. From Figure 3d, the size of the black tissue paper did not change as the temperature gradually rose to 50 °C, while the hydrogel shrank uniformly, which further proved that the pure PNIPAM hydrogel was isotropic and only simple volume contraction/expansion can occur when the temperature changes. In addition, the black tissue paper was also determined to be non-responsive. In addition, as shown in Appendix A, the LCST of pure PNIPAM was close to 34 °C, which was similar to the human body temperature [41]. The macromolecular chain of PNIPAM had both hydrophilic amide groups and hydrophobic isopropyl groups. When the temperature was lower, the amide group can play a dominant role, and the external water molecules can enter the inner molecular chain, which shows the expansion of volume in macroscopic terms. PNIPAM was able to change its volume phase with temperature and convert thermal energy into mechanical energy of motion. So, the actuator can bend or recover to a flat shape with the change of temperature.

The light-responsive (photothermal-responsive) deformations of this bilayer composite actuator were researched with an 808 nm NIR lamp (Figure 4). This anisotropic bilayer structure of the strip-shaped actuator can provide programmable deformations owing to different light-responsiveness levels between the PNIPAM hydrogel layer and the tissue paper layer. In fact, the black tissue paper layer can almost keep the original shape and volume under the irradiation of NIR light and subsequent increase in temperature (Figure 4a), and some of the PNIPAM hydrogel formed in it via in situ polymerization. However, this NIR-heated tissue paper can conduct the heat to the thermal-responsive PNIPAM hydrogel, which can shrink immediately (Figure 3c,d) because the crosslinked PNIPAM network of the hydrogel can become relatively hydrophobic and then discharge most of water in the network to the external environment. As a result, the light-responsive deformations can be achieved owing to the internal stress of the actuator between the shrunk hydrogel layer and the unchanged tissue paper layer [36]. To further corroborate this, we put the composite actuator in 15 °C cold water and irradiated it under NIR light of wavelength 808 nm with intensity 3.5 W/cm^2^, and recorded its temperature change with an infrared thermal imaging instrument. As shown in Figure 4a, the black tissue paper can be heated to about 60 °C in only 90 s, which fully illustrated that the black tissue paper had highly efficient photothermal conversion performance. The combination of black tissue paper and PNIPAM hydrogel made the actuator have an anisotropic bilayer structure, gave it photothermal responsiveness, and made it possible to regulate the deformation of the tissue paper/PNIPAM hydrogel composite actuator by remote control.

Subsequently, we used 3.5 w/cm^2^ NIR light to remotely irradiate the hydrogel composite actuator and study its photothermal-responsive bending performance. Results are shown in Figure 4b, the composite actuator can bend smoothly from 0 to 300° under an irradiated area, which was similar to its thermal-responsive bending process within 45 °C warm water (Appendix A). In addition, as shown in Figure 4c and Appendix A, when the light source was turned off at room temperature, the tissue paper/PNIPAM hydrogel composite actuator could recover its original state in approximately 240 s. It is worth noting that in Appendix A the whole shape bending/recovery process can be repeated at least 30 times almost without any fatigue phenomenon. The actuator can show excellent durability and cycle stability, which was conducive to its further application. Furthermore, the rapid folding process under a stronger point light source was also researched (Figure 4d, Appendix A). Under an NIR point irradiation with 63.9 W/cm^2^ on the middle part of the straight composite actuator, the angle between the two parts of the actuator changed from 0 to 90° in about 2.5 s, which was much faster than its bending speed above because of the stronger light intensity of this point irradiation than that of the area NIR irradiation above. In addition, the folded actuator can also recover fast from 90 to 0° in 120 s (Figure 4e, Appendix A), which was also faster than that of the bended actuator in Figure 4e, which may be because the folding part of the actuator was much smaller than the whole bending part of the actuator. 

### 3.3. Design of Biomimetic Actuating Devices

A biomimetic actuating device was fabricated to simulate the grabbing process of the claw of a hawk (Figure 5a, Appendix A). When the biomimetic “paw” was in cold water at 15 °C, each of its toes had a straight shape. However, under the intensity of 3.5 W/cm^2^ NIR light irradiation, it was gradually heated. As soon as the PNIPAM hydrogel reached its LCST, changing from a hydrophilic state to hydrophobic state, each toe of this “paw” bent and then the object was grabbed firmly in no more than 12 s. In addition, owing to its relatively high-strength mechanical performance, this biomimetic “paw” can grab an object with more than four times its self-weight. Furthermore, another actuating device was also fabricated to simulate the finger deformation of a human hand (Figure 5b, Appendix A). The light-responsive deforming process of the biomimetic “hand” can be controlled by changing the irradiation position of the point light on the “hand”. The folding of each finger of the “hand” can be precisely controlled, which was completed quickly in no more than 2 s because of the much higher intensity of point NIR irradiation than that of the area NIR irradiation above.

## 4. Conclusions

In conclusion, we have successfully fabricated a high-strength tissue paper/PNIPAM hydrogel bilayer composite actuator with printed graphite microparticles based on in situ polymerization for biomimetic light-responsive deformations. The composite actuator can obtain relatively high strength (1.2 MPa of tensile stress) because of the robust tissue paper. Furthermore, owing to the firm combination of the bilayer structure owing to the in situ polymerization of the NIPAM monomers to integrate the tissue paper and hydrogel layer together, the hydrogel composite actuator can realize various controllable deformations. Notably, the graphite microparticles with promising photothermal conversion provide the actuator with NIR light-response performance and remote control, which will further expand the applications of the hydrogel-based biomimetic actuators. Based on the advantages above, two biomimetic actuating devices with remotely controlled light-responsive actuating property have been explored which can simulate the grabbing process of the hawk claw and the precisely controlled finger deformation of the human hand, respectively. This successful design/fabrication of the tissue paper/PNIPAM hydrogel composite actuator provides a new direction for hydrogel-based actuators and will promote the development of other biomimetic intelligent soft actuators.

## Figures and Tables

**Figure 1 polymers-14-05454-f001:**
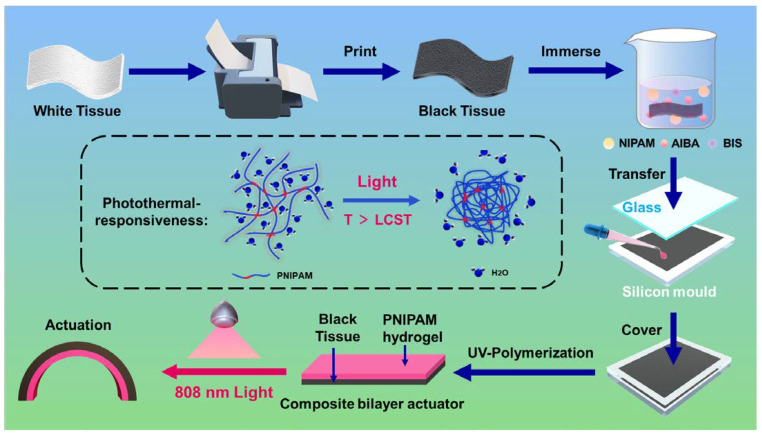
Illustration on fabrication and remotely controlled light-responsive actuation of the tissue paper/PNIPAM hydrogel composite actuator.

**Figure 2 polymers-14-05454-f002:**
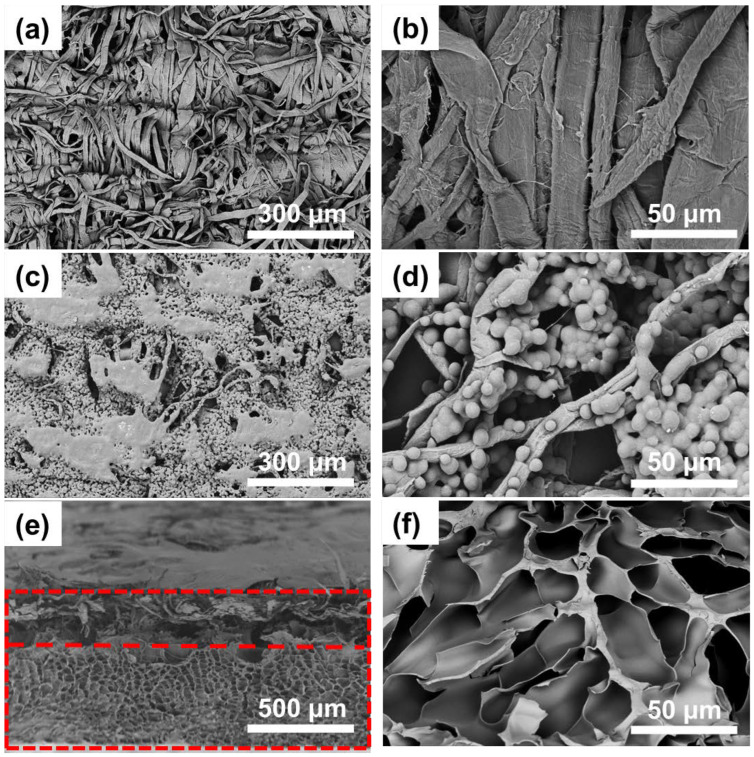
(**a**,**b**) The SEM images of tissue paper. (**c**,**d**) The SEM images of black tissue paper. (**e**) Tissue paper/PNIPAM hydrogel composite actuator structure. (**f**) The SEM image of pure PNIPAM hydrogel.

**Figure 3 polymers-14-05454-f003:**
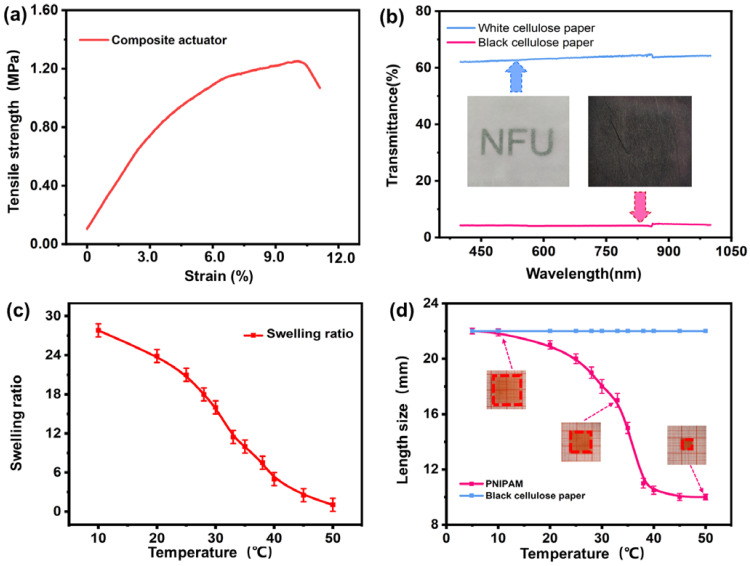
(**a**) The mechanical property of the tissue paper/PNIPAM hydrogel composite actuator. (**b**) Comparison of the light transmittance between the original tissue paper and the black tissue paper dyed by the printer. The (**c**) equilibrium swelling rate and the (**d**) length of the PNIPAM hydrogel at the temperature range from 5 to 50 °C.

**Figure 4 polymers-14-05454-f004:**
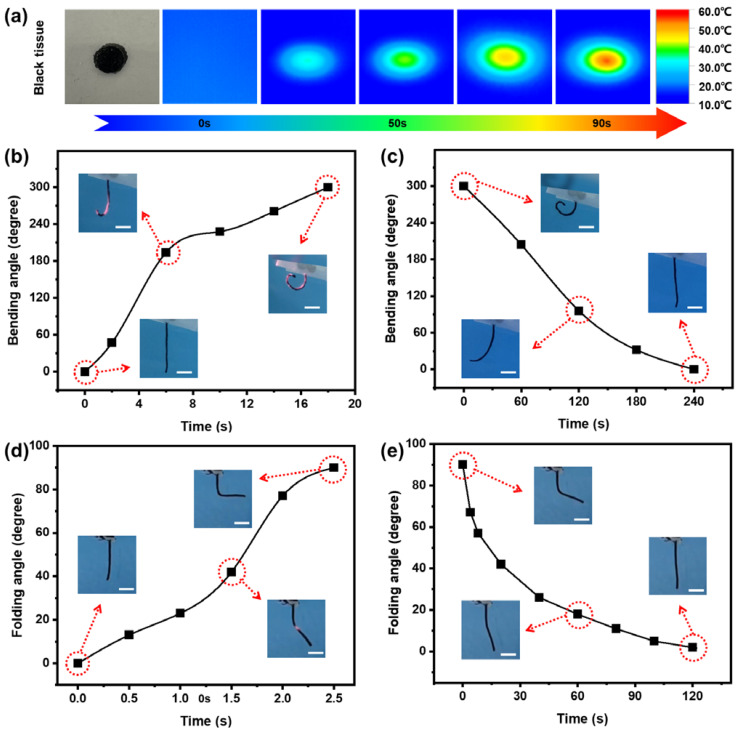
(**a**) Temperature change of tissue paper/PNIPAM hydrogel composite actuator in 15℃ water under 3.5 W/cm^2^ irradiation of the 808 NIR area light. The (**b**) bending (change of central angle, under 3.5 W/cm^2^ irradiation) and the (**c**) recovering process of the tissue paper/PNIPAM hydrogel composite actuator with an area NIR irradiation. The (**d**) folding (change of the angle between the two sides under a 0.2 cm spot diameter and 2 W of power by an 808 nm laser lamp, 63.9 W/cm^2^) and the (**e**) recovering process of the tissue paper/PNIPAM hydrogel composite actuator with a point NIR irradiation.

**Figure 5 polymers-14-05454-f005:**
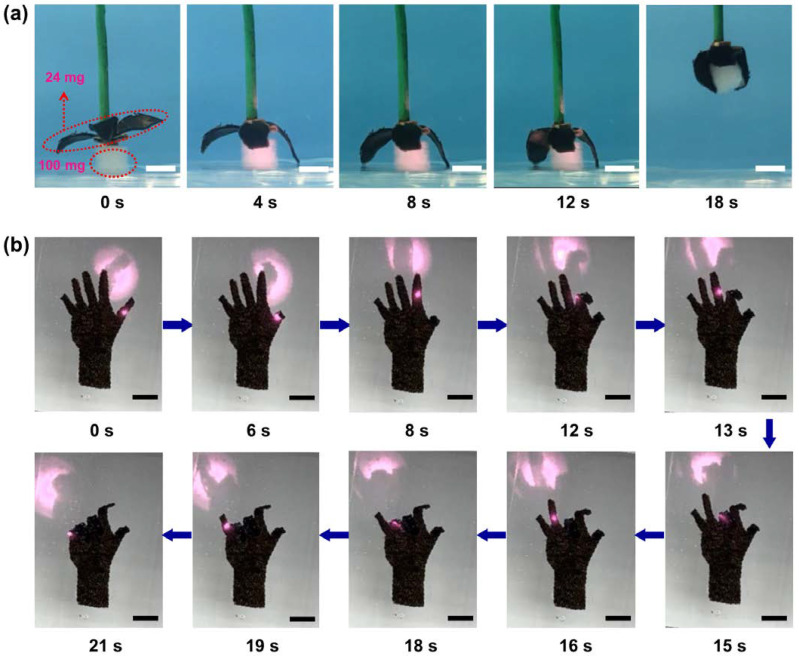
(**a**) The remotely controlled light-responsive grabbing behavior of the biomimetic “paw” based on the composite actuator under an area irradiation (with a 1 cm spot diameter and 5 W of power by an 808 nm laser). (**b**) The remotely controlled light-responsive finger-folding performance of the biomimetic “hand” based on the composite actuator under a point irradiation (with a 0.2 cm spot diameter and 2 W of power by an 808 nm laser).

## Data Availability

Not applicable.

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
