# Peer review of "A Tissue Paper/Hydrogel Composite Light-Responsive Biomimetic Actuator Fabricated by In Situ Polymerization"

_polymers, 2022, doi:10.3390/polym14245454_

Round 1

Reviewer 1 Report

The presented paper describes the fabrication and characterisation of bilayer Paper / PNIPAM hydrogel composites, which display thermally mediated mechanical actuation.  The paper layers are printed with graphite to enable rapid heating. Thermal changes are achieved by the means of NIR irradiation to activate the shape changing process.  The manufacture of the bilayer is well described, and the actuation behaviour is exemplified through a range of designs demonstrating shape changes upon NIR irradiation.

Overall, the work is very interesting and presents a simple, yet effective way to make responsive materials. This is potentially useful, and the demonstrated ability of the construct to grasp an object on demand is a well shown practical application of the material.

Unfortunately, the paper is difficult to follow in places, with the writing and English not making sense.  The writing style and gramma should be reviewed and carefully checked before publication.

I would also encourage the authors to change title of the paper to “A Paper/Hydrogel Composite Light-Responsive biomimetic Actuator Fabricated by In-Situ Polymerization”, and to change references to “tissue” to “paper” or “tissue paper” throughout the text. Referring to the paper phase as tissue is confusing as it gives the impression that the material is of biological origin.  It might also be useful to describe the paper in more detail and expand on how that particular paper was chosen.

Additionally, the anisotropic nature of the bilayer composite is the key to the actuation performance of the material.  The authors should describe this better and describe the nature of the anisotropy with it being achieved through the combination of the bilayer of paper and PNIPAM. It would also be useful to mention the differential thermal expansion of the two layers being the mechanism that gives rise to the shape change.  This discussion is close to describing this concept, but it isn’t explored in detail, and this would be interesting.  For example, have the authors considered looking at other papers which may have different expansion behaviour, or considered looking at different layer thickness for the paper or PNIPAM?

One final comment is on the mechanical performance of the material.  While the addition of the paper layer has clearly had a beneficial effect on the mechanical properties of the material, the modulus achieved is still relatively low in real terms.  A discussion of how this might be increased, or the factors that could influence the modulus achieved would be useful.  If available, it would also be useful to add in data on the force achieved due to actuation.  

Overall, I think the work is interesting and worthy of publication, but the paper is let down by the English which makes it very hard to follow and understand in sections.  I would recommend that the paper is checked and rewritten before resubmission.

Reviewer 2 Report

1. In general, authors should state the meaning of an abbreviation at its first use. For example, lines 56, 86-87, and 107. There are other words in the article, please correct them.

2. In the Materials section, please include the molecular weight of the chemical reagents.

3. Line 136, reference [41] is not appropriate. I recommend replacing it.

4. Line 138, I recommend changing the text: ''highly-enhanced reaching about 1.3 MPa'', it is in contradiction with Figure 3a, where it is clearly seen that it has a value of 1.2 MPa.

5. Line 175, The temperature range is written incorrectly, compared to the scale used in Figure 3d. Please change 10°C to 5°C.

6. Line 186, replace 360° with 300°, in Figure 4a the scale is only up to 300°.

7. Lines 207-209, There are mistakes in the description of figure 4, change b) with d) and c) with e).

8. Line 59, ''Scarpa et al.'' is not found in reference [39]. Line 63, ''Lu et al.'', also does not coincide with the reference [40]. Please correct them.

9. In the Results and Discussion section, the authors did not make a correlation/comparison between the results obtained in this study and the results from the literature. I recommend adding citations.

Round 2

Reviewer 2 Report

The article can be accepted. The authors have significantly improved the article.